# Patients with Non-Alcoholic Fatty Liver Disease and Alcohol Dehydrogenase 1B/Aldehyde Dehydrogenase 2 Mutant Gene Have Higher Values of Serum Alanine Transaminase

**DOI:** 10.3390/jpm13050758

**Published:** 2023-04-28

**Authors:** Tsuo-Hsuan Chien, Chih-Lang Lin, Li-Wei Chen, Cheng-Hung Chien, Ching-Chih Hu

**Affiliations:** 1Department of Gastroenterology and Hepatology, Chang-Gung Memorial Hospital and University, Keelung Branch, Keelung 204, Taiwan; tim1218csm@gmail.com (T.-H.C.);; 2Community Medicine Research Center, Chang-Gung Memorial Hospital and University, Keelung Branch, Keelung 204, Taiwan

**Keywords:** non-alcoholic fatty liver disease, alcohol dehydrogenase, aldehyde dehydrogenase, polymorphism genetic, alleles, ethanol, fatty liver, body mass index

## Abstract

Patients with non-alcoholic fatty liver disease (NAFLD) share similar pathophysiologies to those of patients with alcohol liver disease. Alcoholic metabolic enzyme-related genes (alcohol dehydrogenase 1B (ADH1B) and aldehyde dehydrogenase 2 (ALDH2)) may be associated with pathophysiology in NAFLD patients. In this study, the association between ADH1B/ALDH2 gene polymorphism and serum metabolic factors, body statures, and hepatic steatosis/fibrosis status was evaluated in patients with NAFLD. Using biochemistry data, abdominal ultrasonography, fibrosis evaluation (Kpa), and steatosis evaluation (CAP), ADH1B gene SNP rs1229984 and ALDH2 gene SNP rs671 polymorphism were analyzed in sixty-six patients from 1 January 2022 to 31 December 2022. The percentage of the mutant type (GA + AA) was 87.9% (58/66) in the ADH1B allele and 45.5% (30/66) in the ALDH2 allele. Patients with the mutant-type ADH1B/ALDH2 allele had higher values of alanine aminotransferase (ALT) than the wild type (β = 0.273, *p* = 0.04). No association was observed between body mass index, serum metabolic factors (sugar and lipid profile), CAP, kPa, and ADH1B/ALDH2. A high proportion of the mutant-type ADH1B allele (87.9%) and ALDH2 allele (45.5%) was observed in patients with NAFLD. No association was observed between ADH1B/ALDH2 allele, BMI, and hepatic steatosis/fibrosis. Patients with the mutant-type ADH1B/ALDH2 allele had higher values of ALT than those with the wild type.

## 1. Introduction

Non-alcoholic fatty liver disease (NAFLD) is a common global disease whose prevalence in Asia can reach as high as 40–50% in some regions [1,2]. NALFD and alcohol fatty liver disease share many similar histological features. As such, alcohol metabolic genes, namely alcohol dehydrogenase (ADH) and aldehyde dehydrogenase (ALDH), may be associated with NAFLD [3,4,5,6,7,8]. Endogenous ethanol can be either generated by the intestinal microbiota or be influenced by alcohol metabolic genes [9]. Previous studies have reported that non-alcoholic pediatric patients with non-alcoholic steatohepatitis (NASH) had higher serum levels of ethanol (endogenous ethanol) than obese and healthy children without NASH [9]. The ADH1B (Alcohol Dehydrogenase 1B (Class I), Beta Polypeptide) gene is located on chromosome 4q21–q23. Naturally occurring single nucleotide polymorphisms (SNPs) may be capable of altering ethanol metabolism. The protein encoded by ADH1B gene is a member of the alcohol dehydrogenase family. Members of this enzyme family metabolize a wide variety of substrates, including ethanol, retinol, hydroxysteroids, and lipid peroxidation products [4]. ALDH2 gene is majorly located on chromosome 12 (12q24. 12) and consists of 13 exons with a length of about 44 kilobases [10]. ALDH2 expression could be detected in some organs, such as liver, kidney, lung or stomach [11,12,13]. Among these organs, liver has the highest expression level of ALDH2 [14]. ALDH2 is located in the mitochondrial matrix and functions as three main enzymes, including dehydrogenase, reductase, and esterase. Ethanol metabolism in the liver is mainly oxidized to acetaldehyde (ACH) by alcohol dehydrogenase (ADH). ACH is rapidly converted into acetic acid under the action of acetaldehyde dehydrogenase 2 (ALDH2). Acetic acid is metabolized into carbon dioxide (CO_2_) and water (H_2_O) in peripheral tissues [15]. 4-hydroxynonenal (4-HNE), a toxic aldehyde, can be oxidized to generate 4-hydroxynonenoic acid (4-HNA) by ALDH2 [16]. When the activity of ALDH2 enzyme is reduced, 4-HNE is accumulated which induce a peroxidation. 4-HNE and other toxic aldehydes may lead to DNA and protein dysfunction [17,18]. ALDH2 serves as a tumor suppressor by maintaining the stability of the liver genome [17]. In patients with hepatocellular carcinoma (HCC), the expression level of ALDH2 protein in tumors is significantly lower (down-regulation) than that in normal liver tissues. Hence, ALDH2 is a potential target for the treatment of HCC.

ADH1B gene has two alleles by SNP studies (rs1229984, a G>A base transition in exon 3 leading to the substitution of arginine (Arg48, ADH1B*1) to histidine (His48, ADH1B*2) at position 48). The ADH1B*2 allele encodes for an enzyme with an approximately 80-fold higher turnover rate and 40 times greater activity in producing acetaldehyde than ADH1B Arg48 [19]. ALDH2 is composed of two alleles, producing three genotypes, of which the wild-type GG allele (ALDH2*1/*1) is normally active, the heterozygous mutant AG (ALDH2*1/*2) is approximately one-third as active as ALDH2*1/*1, and the homozygous mutant AA (ALDH2*2/*2) is almost inactive [20,21]. In an analysis of data obtained from the NASH clinical research network, among 1153 patients with biopsy-proven NAFLD, researchers determined that ADH1B*2 reduced the risk of NASH and fibrosis in adults with NAFLD, regardless of their alcohol consumption status [4]. According to one Japanese study, 314 biopsy-confirmed NAFLD patients with a median 7-year follow-up, patients with mutant ALDH2 allele (GA or AA) were associated with a poor prognosis (hazard ratio: 4.568, 95% confidence Interval: 1.294–16.131, *p* = 0.02 by the Cox analysis). They determined that ALDH2 SNP could predict the outcome of NAFLD [22].

Taking into account ethnic differences, a higher proportion (86%) of Asians/Pacific Islanders/Hawaiians have been discovered to carry the ADH1B*2 allele compared to other racial groups (4–13%) [7]. Individuals with the ADH1B*2 allele have a higher alcohol metabolism, which may affect the relationship between moderate alcohol consumption and the histological severity of NAFLD [8]. Furthermore, approximately 40% of East Asian individuals are known to have inherited a mutation of the ALDH2 gene that would induce a decreased efficiency of acetaldehyde metabolism [8,9]. Patients with mutations in the ADH1B/ALDH2 gene may experience facial flushing, nausea, vomiting, and tachycardia after drinking alcohol due to the rapid accumulation of aldehyde. Differences in acetaldehyde metabolism due to mutations in the ALDH2 gene have been previously associated with NASH and esophageal cancer [23,24]. However, studies on the association between hepatic steatosis, fibrosis and the influence of combining ADH1B and ALDH2 allele in patients with NAFLD remain scarce.

This study aimed to elucidate the distribution of ADH1B and ALDH2 alleles in patients with NAFLD. The associations between metabolic factors, BMI, serum alanine transaminase (ALT), histological steatosis or fibrosis, and ADH1B/ALDH2 gene polymorphism were also analyzed. To this end, the following hypotheses were evaluated: (i) the polymorphism of the ADH1B and ALDH2 genes results in different serum metabolites, which influence the development of liver steatosis, inflammation, and fibrosis in patients with NAFLD; (ii) BMI may be associated with ADH1B/ALDH2 in patients of Asian ethnicity with NAFLD.

## 2. Materials and Methods

From 1 January 2022 to 31 December 2022, patients with fatty liver diagnosed by abdominal ultrasonography (US) were enrolled in a ADH1B/ALDH2 gene allele survey at Keelung Chang-Gung Memorial Hospital. The inclusion criteria were an age of ≥20 years and the absence of pregnancy. A standardized questionnaire was administered to all participants. The items in the questionnaire involved alcohol consumption (amount and duration), smoking, and betel nut chewing status. All participants received a demographic survey, a physical examination, and blood tests. The demographic survey assessed the medical history of systemic diseases, such as diabetes mellitus (DM), hypertension, hyperlipidemia, rheumatoid arthritis, autoimmune diseases, and malignancy. Waist girth was measured at the midline between the lowest margin of the subcostal rib and the upper margin of the iliac crest. The exclusion criteria were as follows: (i) men who consumed >30 g of alcohol per day and women who consumed >20 g of alcohol per day; (ii) individuals with secondary causes of steatosis (e.g., corticosteroid use and gastric bypass surgery); (iii) individuals seropositive for hepatitis B surface antigen or anti-hepatitis C virus antibody; (iv) individuals with underlying genetic or metabolic diseases that affect the liver, such as Wilson’s disease, alpha 1-antitrypsin deficiency, genetic hemochromatosis, and autoimmune diseases; (5) individuals who had a blood transfusion within 6 months or a history of bone marrow transplant (to avoid the interference of gene study). The body mass index (kg/m^2^) of the subjects was calculated as the weight (kg) divided by squared height (meter). Subjects were asked to fast overnight before drawing blood samples. Blood tests included serum alcohol concentration, complete blood cell count, aspartate aminotransferase (AST), alanine aminotransferase (ALT), bilirubin and gamma-glutamyl transpeptidase (GGT), fasting sugar and insulin levels, and lipid profiles. ADH1B and ALDH2 gene polymorphism was surveyed. Vibration-controlled transient elastography (VCTE) was performed for liver fibrosis evaluation. This work was approved by The Institutional Review Board of the Chang-Gung Memorial Hospital (IRB no. 202002236B0). All participants agreed to the study conditions and signed the informed consent form before enrollment in this study.

### 2.1. Fatty Liver Evaluation

Liver ultrasonography scanning (Toshiba, Xario, Japan) was performed to assess the degree of fatty liver by two operators who were blinded to the laboratory values. The degree of fatty liver was graded as normal (absent), mild, moderate, or severe according to the basis of intensity, the reflection level of echogenicity (brightness) arising from the hepatic parenchyma with liver–kidney contrast, far attenuation sign by echo penetration into deep portion of the liver, and obscure change in the vessel wall and gallbladder wall [25,26].

### 2.2. ADH1B rs1229984 and ALDH2 rs671 Allele Survey

Genotyping DNA was extracted from leukocytes, and the allele types of ADH1B and ALDH2 were determined using polymerase chain reaction (PCR)–restriction fragment length polymorphism (RFLP). The ADH1B allele tested chromosome 4 rs1229984 SNP and ALDH2 allele tested chromosome 12 rs671 SNP. The reports of ADH1B/ALDH2 genotypes were G/G, G/A, and A/A (allele type *1/*1, *1/*2, and *2/*2, respectively). The genotype of G/G was defined as the ancestral wild type and G/A and A/A were defined as the mutant type (G/A as the heterozygote mutant and A/A as the homozygous mutant). The activity of ADH enzyme in transforming alcohol into aldehyde was lower in individuals with the ADH1B G/G genotype (wild type) than those with the G/A or A/A genotype (mutant type). However, the activity of ALDH enzyme in transforming aldehyde into acetaldehyde was higher in individuals with ALDH2 G/G (wild type) than in those with G/A or A/A (mutant type) [27,28].

### 2.3. Serum Ethanol Measurement

The ethanol concentration was measured immediately after sampling by ETOH2 gen2 (Cobas^®^; Roche, Mannheim, PA, USA). The enzymatic method with alcohol dehydrogenase was used. Ethyl alcohol and nicotinamide adenine dinucleotide (NAD) were converted to acetaldehyde and nicotinamide adenine dinucleotide hydrate (NADH) by ADH. NADH was measured photometrically as a rate of change in absorbance, which was directly proportional to the ethyl alcohol concentration. The measurement range was 2.20–108 mmol/L (10.1–498 mg/dL) [29].

### 2.4. VCTE

FibroScan^®^ (Echosens, Paris, France) is a non-invasive physical tool used for the evaluation of liver fibrosis and steatosis. Technicians trained by Echosens and in possession of the training certificate performed this test. Patients were asked to fast for six hours before examination. Controlled attenuation parameter (CAP) was used as a reference parameter for the diagnosis and monitoring of liver steatosis. The CAP score was measured in decibels per meter (dB/m), ranging from 100 dB/m to 400 dB/m.

The liver stiffness measurement (LSM) result was measured in kilopascals (kPa), which normally ranged between 2 and 7 kPa. In this study, the operators adhered to the following reliability criteria: a ratio of the interquartile range (IQR) to the median (M)(IQR/M) < 0.30 [30].

### 2.5. Statistical Analysis

In a cross-sectional analysis, for continuous variables in a normal (symmetrical) distribution, the values were expressed as the mean ± standard deviation (SD). Otherwise, nonparametric tests were performed. Student’s *t*-test was used to compare the means of two samples, while one-way analysis of variance (ANOVA) was used to compare the mean values of multiple samples. Categorical data were analyzed using the Chi-square test or Fisher’s exact test, as appropriate. Paired sample *t*-test was used for the analysis of repeated measures within the same sample. All statistical tests were two-tailed. *p* < 0.05 was considered statistically significant.

The phi coefficient was applied for binary data (e.g., with or without a fatty liver condition). Spearman’s coefficient Rho was applied as a nonparametric measure of rank correlation, and Pearson’s correlation was applied for continuous data. Confounding factors, such as age, gender, and BMI, were adjusted in the regression analysis. The adjusted odds ratio (aOR) was recorded with 95% confidence interval (CI). The variables were genotypes of ADH1B and ALDH2, liver steatosis (CAP), and fibrosis severity (kPa). The distribution of the ADH1B and ALDH2 genotypes, liver steatosis, and fibrosis severity were recorded.

Regression analyses were performed between liver steatosis/fibrosis and ADH and ALDH in each group. Statistical analyses were performed using PASW for Windows (version 18.0) (SPSS Inc., Chicago, IL, USA).

## 3. Results

Initially, 176 patients were enrolled in the study. After the questionnaire was administered, 108 patients were excluded by exclusion criteria. Two patients were excluded for missing data. Finally, 66 participants were included in the analysis (Figure 1).

Table 1 shows the demography of the participants, including the ADH1B and ALDH2 genes (wild type and mutant type). The mean age was 54.7 ± 12.4 years old. Forty participants were women (40/66, 60.6%). The percentage of mutant type (GA + AA) was 87.9% (58/66) in the ADH1B allele and 45.5% (30/66) in the ALDH2 gene allele. The gender distribution differed for the ALDH2 allele, with more females (26/40, 65%) present in the ALDH2 wild-type group. The serum ethyl ethanol concentration was <10 mg/dL in all participants. According to the BMI values, 27 participants (40.9%) were lean (BMI < 23 kg/m^2^). The distribution of ADH1B and ALDH2 gene polymorphism was not statistically different between the lean and obese groups. No association was observed between the ADH1B/ALDH2 allele and BMI values. The mean values of total cholesterol and LDL were higher in participants with the ADH1B mutant type than in those with the wild type (total cholesterol: 184.3 ± 34.6 vs. 155.1 ± 32.3 mg/dL, *p* = 0.028; LDL: 115.1 ± 35.5 vs. 81.6 ± 19.5 mg/dL, *p* = 0.028). The mean value of ALT was higher in the ALDH2 gene mutant-type group than in the wild-type group (44.3 ± 36.7 vs. 28.3 ± 17.4 IU/L, *p* = 0.037).

Table 2 shows that ALT was associated with the ADH1B/ALDH2 allele. Participants were divided into three groups according to the ADH1B and ALDH2 allele. Participants in group 1 had both wild-type genes (ADH1B and ALDH2) (GG type, n = 6) and were used as the reference group. Participants in group 2 had one wild-type (GG) gene and one mutant-type (GA or AA) gene (n = 32). Participants in group 3 had both mutant-type (GA or AA) genes (n = 28). The ALT values (mean ± standard deviation) of the participants were 27.4 ± 15.4 IU/L, 31.5 ± 25.8 IU/L, 46.0 ± 37.5 IU/L in group 1, group 2 and group 3. A positive correlation was detected for ALT values in these 3 groups by using group 1 as reference (β = 0.273, *p* = 0.04).

Table 3 shows the relationship between the ADH1B/ALDH2 alleles and participant age, gender, BMI, and hepatic steatosis/fibrosis. BMI was determined to be positively correlated with hepatic steatosis (CAP values) and fibrosis (kPa values) (*p* < 0.05). The allele status (mutant type or wild type) was not correlated with CAP or kPa after adjusting for the confounding factors of age, gender, and BMI. Furthermore, no association was observed between isolated ADH1B allele or isolated ALDH2 allele and CAP or kPa (Appendix A).

Table 4 revealed AHD1B/ALDH2 allele was not correlated with BMI after adjusting the factors of age and gender.

## 4. Discussion

In this study, participants with NAFLD and either ADH1B or ALDH2 mutant alleles had higher ALT values than those with the corresponding wild-type alleles. The higher ALT values observed in the participants with mutant alleles are compatible with those reported in previous studies [31,32]. For example, in a study evaluating a Japanese cohort, among 341 patients who never or seldom drank for 4–6 years, the incidence of NAFLD was determined to be higher in individuals who carried the mutant gene ALDH2*2 compared to non-carriers [31]. Another study reported that the BMI, TG level, and ALDH2 genotype were associated with ALT elevation [32]. However, Vilar-Gomez et al. discovered that patients with NAFLD and ADH1B*2 (mutant allele) had a decreased risk of steatohepatitis and fibrosis, and this protective effect remained significant even after controlling for alcohol consumption status, age, sex, and BMI. Hence, they concluded that ADH1B*2 reduces the risk of NASH and fibrosis in adults with NAFLD, regardless of their alcohol consumption status [4]. Another randomization study also determined that carriers of the ADH1B*2 allele had not only lower rates of alcohol consumption but also decreased scores of histologic steatosis, lobular inflammation, and NAFLD activity. The authors assumed that the effect of ADH1B*2 on the liver histology was mediated exclusively by the amount of alcohol consumed [33]. Other metabolites, such as retinoic acid or 4-HNE, may be involved in liver inflammation in patients with ADH1B/ALDH2 mutant alleles [34,35].

ADHs is involved in non-alcohol-related molecular pathways, including all-trans-retinol and retinoic acid [34,36]. A significant accumulation of the 4-HNE protein and an upregulation of ALDH2 protein have been reported in patients with NASH. 4-HNE is considered a biomarker of oxidative stress and is identified as one of the most formidable reactive aldehydes [35,37]. Mutations in the ALDH2 gene have been found to reduce the oxidative stress protection response of liver recovery in NASH patients [35]. However, further study is warranted to elucidate the metabolic substances involved in the mechanism of liver inflammation in non-alcohol drinking patients with NAFLD and ADH1B/ALDH2 mutant alleles.

In the present study, BMI was significantly associated with factors of hepatic steatosis (CAP) and fibrosis (kPa). Although the mean values of BMI were higher in the participants of the mutant group, no statistical significance was determined. This may be due to a small sample number in this study. Although most patients with fatty liver are overweight or obese, approximately 20% of patients with NAFLD in Asia are not obese (BMI < 23 kg/m^2^) [38]. Ethnic discrepancies and genetic polymorphism may explain fatty liver formation in these lean patients [38,39]. In previous studies conducted in China and East Asia, the ALDH2 mutant allele was significantly associated with obesity, increased BMI, and visceral fat deposition [40,41]. Individuals with the ADH1B*2 allele were reported to have a more rapid ethanol metabolism and elimination. As such, the use of ethanol as a source of energy is less efficient in these individuals.

Those who consume excessive amounts of alcohol and have the ADH1B*2 allele may experience lower weight gain if no other food except alcohol is consumed. A differential effect of the ADH1B*2 allele on body weight was based on alcohol consumption status [42]. Among patients with a history of moderate alcohol consumption, body weight was determined to be significantly lower in carriers of ADH1B*2 compared with ADH1B*1; however, no effect was seen among non-drinkers [42]. In the present study, no association was observed between the ADH1B allele and BMI. This may be due to the fact that the majority of our participants were non-drinkers. Since the participants in this study were not consuming excessive amounts of alcohol and were asked to fast for over 12 h, the serum alcohol concentration was less than 10 mg/dL in all participants. However, food consumed in daily meals can also contain alcohol. For example, wine is a common ingredient in cooking. Furthermore, the consumption of certain types of cookies and fruits, such as egg yolk pie, fermented tofu, litchi, or Durian, has been reported to increase the ethyl concentration in breath samples [43]. Thus, participants with mutations in the ADH1B and ALDH2 gene may have a higher baseline serum ethyl or aldehyde concentration when consuming food containing wine or alcohol. However, the incidence of higher serum ethyl or aldehyde levels could not be confirmed in the present study because the participants were asked to fast and avoid foods containing alcohol before being evaluated. Another possibility is the presence of non-dietary ethanol (endogenous alcohol) derived from gastrointestinal tract bacteria [9,44]. Endogenous alcohol can induce cytochrome P450 2E1 enzyme upregulation, which catalyzes the oxidation of ethanol. However, the production of free radicals results in oxidative damage, mitochondrial dysfunction, and liver inflammation [45]. The influence of food on the development for NAFLD or NASH may be different in people with different ALDH2 allele. In a Chinese study, salted and smoked food intake was a factor associated with a higher probability of having NAFLD or NASH in the mutant ALDH2 genotype, but there was no effect in the wild ALDH2 genotype [46].

This study has several limitations. First, the size of the study sample was small (<100). Second, the levels of serum aldehyde and oxidative metabolite, such as 4-HNE, were not evaluated. Third, no endogenous alcohol or gut microbiome data were evaluated in this study. Thus, further studies will be needed to elucidate the mechanism underlying the elevated ALT levels observed in patients with NAFLD and mutant ADH1B/ALDH2 alleles.

## 5. Conclusions

In summary, a high proportion of mutations in the ADH1B gene (87.9%) and ALDH2 gene (45.5%) was observed in this study. Higher ALT values were observed in patients with NAFLD and ADH1B/ALDH2 mutant alleles. Furthermore, no association was observed between ADH1B/ALDH2 mutant alleles and hepatic steatosis/fibrosis.

## Figures and Tables

**Figure 1 jpm-13-00758-f001:**
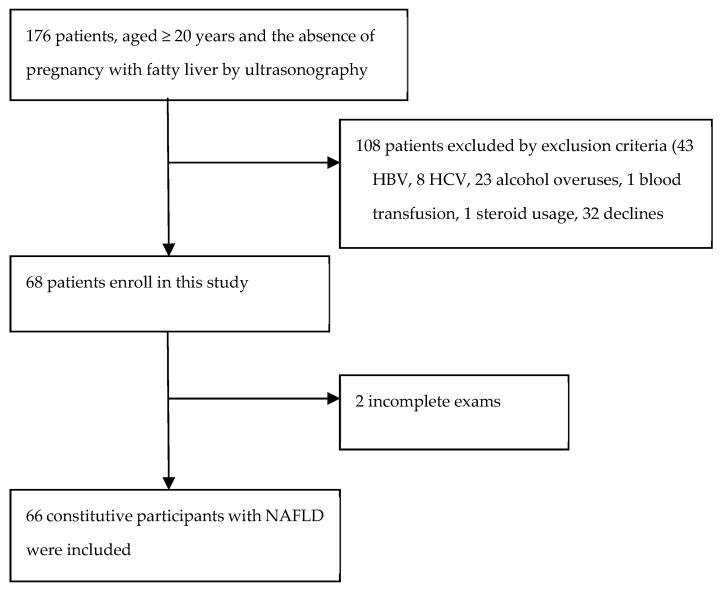
Flow chart for patients enrolled in a ADH1B/ALDH2 gene allele survey at Keelung Chang-Gung Memorial Hospital.

**Table 1 jpm-13-00758-t001:** Demography of participants.

	Total	ADH1B		*p*-Value	ALDH2		*p*-Value
		Mutant	Wild		Mutant	Wild	
Number	66	58	8		30	36	
Age †	54.5 (±12.4)	54.3 (±12.7)	56.2 (±10.5)	0.692	52.9 (±12.2)	56.0 (±12.6)	0.319
**Gender**							
Male	26 (39.4%)	24 (36.4%)	2 (3.0%)	0.376	16 (24.2%)	10 (15.2%)	0.037 *
Female	40 (60.6%)	34 (51.5%)	6 (9.1%)		14 (21.2%)	26 (39.4%)	
**BMI ^†^**	25.1 (±4.4)	25.2 (±4.5)	24.2 (±3.4)	0.563	25.4 (±4.4)	24.9 (±4.4)	0.649
**Fibroscan**							
kPa ^‡^	6.3 (±4.8)	6.5 (±5.1)	4.6 (±1.2)	0.307	6.6 (±4.5)	6.0 (±5.1)	0.603
CAP ^&^	279.1 (±56.3)	280.0 (±57.7)	272.3 (±48.2)	0.721	286.3 (±62.5)	273.0 (±50.9)	0.352
**Blood Sugar**							
Glucose	108.3 (±30.3)	107.7 (±31.1)	112.8 (±24.6)	0.657	106.8 (±37.5)	109.5 (±23.7)	0.727
HbA1c	6.0 (±0.9)	5.9 (±0.9)	6.3 (±0.9)	0.282	5.9 (±1.0)	6.0 (±0.8)	0.562
**Lipid profile**							
TCHOL ^$^	180.6 (±35.5)	184.3 (±34.6)	155.1 (±32.3)	0.028 *	182.8 (±31.1)	178.9 (±38.9)	0.665
TG ^#^	133.3 (±69.8)	133.6 (±71.3)	131.3 (±62.7)	0.932	129.5 (±56.4)	136.3 (±79.4)	0.704
HDL-C ^₤^	54.9 (±14.5)	55.1 (±15.0)	52.6 (±10.2)	0.692	52.9 (±13.3)	56.6 (±15.6)	0.346
LDL-C €	111.5 (±35.5)	115.1 (±35.4)	81.6 (±19.5)	0.028 *	112.0 (±27.1)	111.0 (±41.9)	0.914
**Liver Tests**							
AST ^∫^	27.3 (±11.8)	27.5 (±11.9)	25.5 (±11.8)	0.645	30.1 (±11.9)	24.9 (±11.4)	0.077
ALT ^∆^	35.6 (±28.8)	36.6 (±29.6)	28.6 (±22.4)	0.464	44.3 (±36.7)	28.5 (±17.4)	0.037 *
Alkp ^□^	74.8 (±19.1)	74.2 (±18.8)	80.6 (±21.9)	0.438	71.4 (±13.6)	77.5 (±22.3)	0.228
BilT ^◊^	0.6 (±0.3)	0.6 (±0.3)	0.5 (±0.1)	0.183	0.7 (±0.4)	0.6 (±0.2)	0.320
GGT ^●^	40.0 (±58.6)	42.2 (±62.2)	24.5 (±8.5)	0.428	36.3 (±29.2)	43.1 (±75.2)	0.643

† BMI: body mass index; ‡ kPa: kilopascals; & CAP: controlled attenuation parameter, dB/m; $ TCHOL: total cholesterol, # TG: triglyceride; ₤ HDL-C: high density of cholesterol; € LDL-C: low density of cholesterol; ∫ AST: aspartate transaminase; ∆ ALT: alanine transaminase; □ Alkp: alkaline phosphatase; ◊ BILT: total bilirubin; ● GGT: gamma glutamine transpeptidase. * The mean values were higher in participants with the ADH1B mutant type than in those with the wild type (*p* < 0.05).

**Table 2 jpm-13-00758-t002:** Correlation analysis between ALT and ADH1B/ALDH2 allele.

Model		Unstandardized Coefficients		Standardized Coefficients	T	Sig.
		B	Std. Error	β		
ALT	(Constant)	72.447	15.838		4.574	0.000
	3 groups	18.721	8.888	0.273	2.106	0.040

Note: Dependent Variable: ALT (R = 0.273. R^2^ = 0.075. Adj. R^2^ = 0.058). Group 1: ADH1B + ALDH2 (wild) (reference group); Group 2: ADH1B (mutant) + ALDH2 (wild) or ADH1B (wild) + ALDH2 (mutant); Group 3: ADH1B + ALDH2 (mutant).

**Table 3 jpm-13-00758-t003:** Correlation analysis between CAP, kPa and ADH1B/ALDH2 allele.

Model		Unstandardized Coefficients		Standardized Coefficients		Sig
		B	Std. Error	β	T	
CAP	(Constant)	138.768	51.770		2.680	0.010
	3 groups	−2.520	9.677	−0.029	−0.260	0.795
	Age	−0.038	0.537	−0.008	−0.072	0.943
	Gender	−12.693	13.530	−0.111	−0.938	0.352
	BMI	6.646	1.426	0.523	4.661	0.000
kPa	(Constant)	−9.954	4.852		−2.052	0.045
	3 groups	−1.012	0.907	−0.134	−1.116	0.269
	Age	0.107	0.050	0.254	2.122	0.038
	Gender	0.829	1.268	0.083	0.653	0.516
	BMI	0.425	0.134	0.382	3.183	0.002

Note: 1. Dependent Variable: CAP (R = 0.566. R^2^ = 0.321. Adj. R^2^ = 0.274). kPa (R = 0.472. R^2^ = 0.223. Adj. R^2^ = 0.169). 2. adjusting factors of age, gender and BMI.

**Table 4 jpm-13-00758-t004:** Correlation analysis between BMI and ADH1B or ALDH2 allele.

	Unstandardized		Standardized Coefficients		
BMI	B	Std. Error	β	T	Sig
Constant	29.944	3.135		9.551	0.000
ADH1B allele	−0.598	1.655	−0.044	−0.361	0.719
Age	0.004	0.046	0.010	0.081	0.936
Gender	−2.670	1.170	−0.294	−2.282	0.026
Constant	29.269	2.925		10.0	0.000
ALDH2 allele	0.084	1.125	0.009	0.075	0.941
Age	0.003	0.046	0.009	0.068	0.946
Gender	−2.728	1.194	−0.300	−2.284	0.026

Note: 1. Dependent Variable: BMI. 2. adjusting factors of age, gender.

## Data Availability

All data are present within the article and can be further obtained from the corresponding author on reasonable request.

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
