# Peer review of "Patients with Non-Alcoholic Fatty Liver Disease and Alcohol Dehydrogenase 1B/Aldehyde Dehydrogenase 2 Mutant Gene Have Higher Values of Serum Alanine Transaminase"

_jpm, 2023, doi:10.3390/jpm13050758_

Round 1

Reviewer 1 Report

The work titled: "Patients with non-alcoholic fatty liver disease and alcohol dehydrogenase 1B/aldehyde dehydrogenase 2 mutant genes have higher values of serum alanine transaminase" presents a characterization study of the alleles present for ADH1B and ALDH2, ancestral and mutant and correlates it with the non-alcoholic fatty liver disease.

The work is well presented, and the data is simple but straightforward. However, the frequency relationship of the mutated alleles between patients and the open population corresponding to the population studied is unclear, so the strength of their proposed correlation is diluted. It is recommended to clarify this point.

In 2.2. I understand that the G/G allele for both ADH and ALDH is the wild-type or ancestral homozygous, G/A is the heterozygote for either, and A/A is the mutated homozygous for either.

Delete section 0 is required.

Table 2 titles have an error.

Line 40 "live" or "liver"

No comments, errors are minor, and even printing.

Author Response

Point 1: The work titled: "Patients with non-alcoholic fatty liver disease and alcohol dehydrogenase 1B/aldehyde dehydrogenase 2 mutant genes have higher values of serum alanine transaminase" presents a characterization study of the alleles present for ADH1B and ALDH2, ancestral and mutant and correlates it with the non-alcoholic fatty liver disease.

The work is well presented, and the data is simple but straightforward. However, the frequency relationship of the mutated alleles between patients and the open population corresponding to the population studied is unclear, so the strength of their proposed correlation is diluted. It is recommended to clarify this point.

In 2.2. I understand that the G/G allele for both ADH and ALDH is the wild-type or ancestral homozygous, G/A is the heterozygote for either, and A/A is the mutated homozygous for either.

Response 1: According to one review article, the frquency of mutated alleles for ADH1B Genotypes in han Chinese and Taiwanese was (91%) and for ALDH2 Genotypes was (35%). This study may has similar result compare with general population. Thus, the porportion of G/A allele and A/A alleles was not analysis in this study. It is also one of limitation of this study.

Point 2:

Delete section 0 is required.

Table 2 titles have an error.

Line 40 "live" or "liver"

Response 2: This erro has been correct in updat. Thanks a lot!

Reviewer 2 Report

Authors elucidated the possible distribution of the ADH1B/ALDH2 gene mutations among NAFLD patients.  As authors mentioned in the discussion part, this study have several limitations. However, as a preliminary study to evaluate the association of the ADH1B/ALDH2 gene mutations with pathological features of NAFLD, this study may provide some information to carry out further detailed studies to delineate the mechanism underlying this casual association. I suggest few minor issues to be fixed before publishing this study in the Journal of Personalized Medicine.

1.     Table 2 should be modified to show the actual ALT values in each group. Such as ALT value in group 1 both wild-type genes, group 2 one wild-type (GG) gene and one mutant-type (GA or AA) gene and in group 3 both mutant-type (GA or AA) genes.

2.     It will be interesting if authors discuss the possible use of these polymorphisms to predict NAFLD outcome or these polymorphisms as biomarkers.

3.     Introduction section can be improved by adding background information about the ADH1B and ALDH2 genes like what are these genes and its functions, any known factors regulating the expression of these genes, its role in alcohol metabolism, etc.

4.     There are some typo errors in the manuscript. e.g. SNP rs1229984 was mentioned as s1229984 in abstract; line 17.

5.     Section 0. How to use this template should be removed; lines 30-36.

Author Response

Point 1: Table 2 should be modified to show the actual ALT values in each group. Such as ALT value in group 1 both wild-type genes, group 2 one wild-type (GG) gene and one mutant-type (GA or AA) gene and in group 3 both mutant-type (GA or AA) genes.

Response 1: ALT values in each group in group 1 and group 2 will be add in table 2 in next update.

Point 2: It will be interesting if authors discuss the possible use of these polymorphisms to predict NAFLD outcome or these polymorphisms as biomarkers.

Response 2: In this study, the elevation of ALT was noted in group 2 (one wild-type (GG) gene and one mutant-type (GA or AA) but there was no significant(P > 0.05). Only Group 3 has a siginifcant elevated ALT (P = 0.04). These results should be validated in larger sample base studies.

Point 3:  Introduction section can be improved by adding background information about the ADH1B and ALDH2 genes like what are these genes and its functions, any known factors regulating the expression of these genes, its role in alcohol metabolism, etc.

Response 3: I will improve background informaion in introducation section in next update.

Point 4: There are some typo errors in the manuscript. e.g. SNP rs1229984 was mentioned as s1229984 in abstract; line 17.

 Response 4: This erro has been correct in updat. Thanks a lot!

Point 5.     Section 0. How to use this template should be removed; lines 30-36.

Response 5: This erro has been correct in updat. Thanks a lot!